# Photocatalytic Degradation and Mineralization of Estriol (E3) Hormone Using Boron-Doped TiO$_2$ Catalyst

**Laura Yanneth Ramírez-Quintanilla [1], Diego Pino-Sandoval [1], Juan Camilo Murillo-Sierra [2], Jorge Luis Guzmán-Mar [1], Edgar J. Ruiz-Ruiz [1,\*] and Aracely Hernández-Ramírez [1,\*]**

[1] Facultad de Ciencias Químicas, Universidad Autónoma de Nuevo León, Ave. Universidad s/n, Cd. Universitaria, San Nicolás de los Garza C.P. 66455, Nuevo León, Mexico

[2] Facultad de Ciencias Químicas, Universidad de Concepción, Edmundo Larenas 129, Concepción C.P. 4070386, Chile

\* Correspondence: edgar.ruizrz@uanl.edu.mx (E.J.R.-R.); aracely.hernandezrm@uanl.edu.mx (A.H.-R.)

**Abstract:** In this research work, boron-doped titanium oxide (B-TiO$_2$) was prepared by the sol-gel method to investigate its behavior in the degradation of the recalcitrant hormone estriol (E3). The doped photocatalyst was synthesized at different boron/titania ratios of 2, 3, and 5 wt.% of boron with respect to the TiO$_2$ content. The obtained materials were characterized by UV-Vis diffuse reflectance spectroscopy (DRS), X-ray diffraction (XRD), Raman spectroscopy, Scanning electron microscopy (SEM), and X-ray photoelectron spectroscopy (XPS). The textural properties, specific surface area, and porosity were obtained from N$_2$ adsorption–desorption isotherms by BET and BJH methods, respectively. The photocatalytic performance of each synthesized catalyst was evaluated on the degradation of an aqueous solution (10 mg/L) of estriol (E3) under simulated solar radiation. The variation in the hormone concentration was determined by the HPLC technique, and the mineralization was evaluated by the quantification of total organic carbon (TOC). The obtained results indicated that the catalyst with 3 wt.% of boron incorporation exhibited the best performance on the degradation and mineralization of estriol, achieving its complete degradation at 300 kJ/m$^2$ of accumulated energy and 71% of mineralization at 400 kJ/m$^2$ (2 h) obtaining a non-toxic effluent.

**Keywords:** heterogeneous photocatalysis; sol-gel method; B-TiO$_2$; estriol

## 1. Introduction

In recent decades, endocrine-disrupting compounds (EDCs) have been detected in drinking water and aquatic systems all over the world [1–4]. Steroids that affect aquatic life due to their potential for endocrine disruption are mainly natural and synthetic estrogens and contraceptives, which include estrone (E1), 17β-estradiol (E2), estriol (E3), 17α-ethinylestradiol (EE2), and mestranol (MeEE2) [5,6]. The residual estriol (E3) in the aquatic environment has raised extensive attention for the higher estrogenic activity, even at low concentrations [7,8]. Estriol ((16alfa, 17beta)-estra-1, 3,5(10)-trieno-3, 16,17-triol) is the primary estrogen found in urine, it is a byproduct of the metabolism of estrone in non-pregnant women, and in the case of pregnancy, the placenta is the main source of estriol [9]. E3 replaces the lack of estrogenic production in menopausal women and is a synthetic oral contraceptive, so it is used as a pharmaceutical product.

The low efficiency in the removal of this type of contaminants by conventional processes in water treatment plants has resulted in their permanence in wastewater and drinking water. Several processes have been used to treat estriol-contaminated water, including microbiological [10], electrochemical [11], ozone [12], and photocatalysis [13].

Heterogeneous photocatalysis, by using TiO$_2$ for the pollutants' oxidation, has attracted considerable attention in water treatment applications. However, due to wide band gap (3.2 eV) of TiO$_2$, its photoactivity is limited to UV irradiation [14]. Another drawback that restricts the application of TiO$_2$ is the fast recombination rate of the photogenerated



$e^-/h^+$ pairs. In order to use solar energy and decrease the charge carrier's recombination, many studies on non-metal doping have been carried out to extend the spectral response of $TiO_2$ into the visible region and enhance its photocatalytic behavior. Some studies related to doping $TiO_2$ with N [15], C [16], S [17], P [18], and B [19–21] have demonstrated that the insertion of dopant impurities into the titania structure affects the electronic band edges or introduces impurity states in the band gap [22]. The increase in the photocatalytic activity can be related to a lower $e^-/h^+$ recombination and light absorption in a large range of the spectrum. Several authors reported that doping with boron into $TiO_2$ resulted in the improvement of $TiO_2$ photocatalytic efficiency, which may be related to a decrease in the band gap energy (Eg) and an increase in the light absorption range [20,21,23,24].

Quiñones et al. [25] investigated different boron-doped $TiO_2$ photocatalysts that were synthesized by the sol-gel method and used in the photocatalytic treatments for four pesticides. The boron was incorporated into interstitial positions of $TiO_2$ and did not modify the band gap energy with respect to bare $TiO_2$.

Wang et al. [26] synthesized a boron or cerium single-doped $TiO_2$ and B-Ce co-doped $TiO_2$ using a modified sol-gel method. The catalysts exhibited excellent photocatalytic antimicrobial activity. B and Ce co-doping decreases the band gap of the $TiO_2$ from 3.19 to 2.68 eV and effectively inhibits $e^-/h^+$ recombination, which facilitates the absorption of visible light. The substitutional B occupying O sites and interstitial B atoms result in the generation of impurity states and oxygen vacancies and promote the conversion of Ti from Ti(IV) to Ti(III). Higher activity was observed with co-doped materials because of the narrow band gap, an increase in the specific surface area, and surface-active oxygen species.

Wang et al. [27] synthesized boron-doped (B-doped) $TiO_2$ with a tunable anatase/rutile ratio for efficient atrazine degradation by using a simple one-step calcination method. The prepared B-doped $TiO_2$ exhibited a higher photocatalytic activity for atrazine degradation, showing a reaction rate four times faster than that of the non-doped counterpart. Compared with the non-doped $TiO_2$, the B-doped $TiO_2$ (A/R) could greatly increase the photocatalytic oxidation activity of $TiO_2$ by positively shifting the valence band energy level, which favored the $^\bullet$OH radical formation [19].

Bilgin et al. [28] synthesized boron-doped $TiO_2$ (B-$TiO_2$) photocatalysts with a visible light activity using the solvothermal method. The obtained B-doped $TiO_2$ showed high activity for endocrine-disrupting compounds (bisphenol-A, 2, 4-dichlorophenol) and non-steroidal anti-inflammatory drugs (ibuprofen, flurbiprofen) oxidation under UV-A and visible irradiation ascribed to a decrease in the band gap energy and a more developed surface area.

All the outcomes mentioned above enable the B-doped $TiO_2$ to be an efficient photocatalyst. Nevertheless, although several studies related to B-$TiO_2$ catalysts for photocatalytic applications have been reported [20,27,29,30], the content of boron, the $TiO_2$ precursors, and the calcination temperature define the structural, optical, and textural properties of the materials and these, in turn, affect their photocatalytic behavior. Therefore, the effect of $TiO_2$ boron doping still deserves to be further investigated since its behavior also depends on the used radiation source due to different results have been obtained regarding the contribution of B on the $TiO_2$ bandgap energy. Additionally, until our knowledge, the doped B-$TiO_2$ material has not been explored on the photocatalytic degradation reaction of the E3 hormone in aqueous media.

Therefore, in this study, the photodegradation of the endocrine-disrupting compound estriol (E3) using the B-doped $TiO_2$ catalysts (2, 3, and 5 wt.% of boron content) under simulated solar light was investigated. The degradation and mineralization of the hormone molecule were evaluated at different catalyst loadings (0.5, 1.0, and 1.5 g/L), and the toxicity evolution during the photocatalytic process using a *Vibrio fischeri* bioassay was assessed.

## 2. Results and Discussion

### 2.1. Photocatalysts Characterization

Figure 1a depicts the XRD patterns of B-TiO$_2$ with different boron amounts and pure TiO$_2$ powders. Reflections that are attributed to the anatase TiO$_2$ crystalline phase are detectable in pure TiO$_2$ and B-TiO$_2$ (JCPDS 21-1272). The XRD patterns show the characteristic peaks at a 25.5°, 38°, 48.1°, 54.2°, 55°, 63°, 70°, and 75° 2θ angle corresponding to anatase form. Nevertheless, the addition of boron induces TiO$_2$ transformation to the rutile phase since the doped samples show an extra peak at a 2θ angle of 27.5°, which is attributed to the most intense reflection of the rutile plane (110). This result indicates that B was incorporated into the structure of TiO$_2$ [21]. Through the Scherrer formula (D = 0.89λ/βcosθ), the crystallite sizes of the TiO$_2$ and B-doped TiO$_2$ samples were calculated. As can be observed in Table 1, the crystallite size decreases as the boron content increases. These results are consistent with the studies of Cavalcante et al. [31] and Khan et al. [32], who reported that this effect could be explained in terms of the interstitial B$^{3+}$ ions that balance the residual charge of TiO$_2$ and lead to a reduction in surface energies, which in turn impede the grain growth.

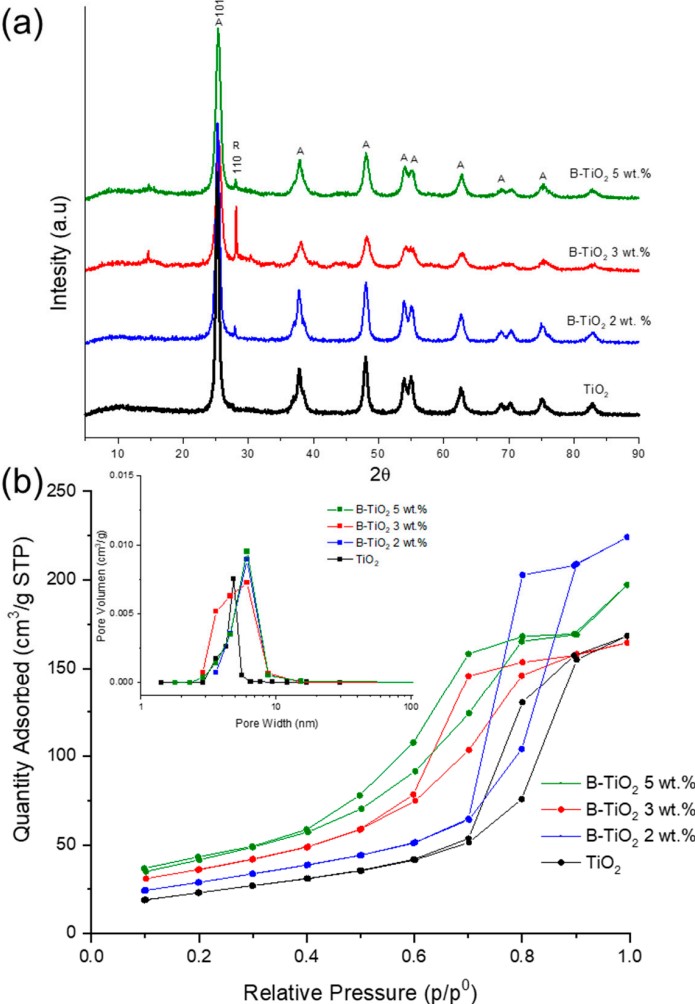

**Figure 1.** XRD patterns and (**a**) N$_2$ adsorption–desorption isotherms of (**b**) Boron-doped and non-doped TiO$_2$ photocatalysts.

The recorded N$_2$ physisorption isotherms of the prepared materials are depicted in Figure 1b. The observed isotherms are classified as type IV according to IUPAC classification, and they are characteristic of mesoporous solids. The characteristic H1-type hysteresis loops are distinctive of powders that possess ink-bottle-type pores.

The specific surface area of the photocatalysts were calculated from the adsorption–desorption isotherms by the Brunauer-Emmett-Teller (BET) method, and the values are shown in Table 1. Compared with pure $TiO_2$, the B-doped $TiO_2$ sample shows a high specific surface area, which is consistent with the small crystallite size obtained by XRD. In general, a semiconductor with a larger surface area tends to have a better capacity for pollutant adsorption, improving its photocatalytic activity [33]. The average pore size values obtained from 4.88 to 6.12 nm, as seen in the inset of Figure 1b, belong to mesoporous materials (2–50 nm).

**Table 1.** Characterization results for $TiO_2$ and B-$TiO_2$ catalysts.

| Catalysts | [a] Crystallite Size (nm) | $S_{BET}$ (m²/g) | [b] Average Pore Size (nm) | Eg (eV) | [c] B Incorporated (wt. %) |
|---|---|---|---|---|---|
| $TiO_2$ | 17.81 | 85 | 4.88 | 3.09 | n.d |
| B-$TiO_2$ 2 wt.% | 9.13 | 121.3 | 6.11 | 3.13 | $1.3 \pm 0.05$ |
| B-$TiO_2$ 3 wt.% | 7.33 | 153.2 | 6.11 | 3.06 | $2.7 \pm 0.03$ |
| B-$TiO_2$ 5 wt.% | 6.06 | 130.7 | 6.12 | 3.13 | $2.2 \pm 0.04$ |

[a] Calculated by the Scherrer equation. [b] Calculated by BJH method. [c] Carminic acid assay.

The UV–Vis-DRS spectra were recorded to determine the band gap values of the synthesized catalysts. In Figure 2a, it can be observed that the absorption edges of the doped materials are comparable to that of pure $TiO_2$. The Kubelka–Munk function was used to transform the measured reflectance spectra into the corresponding absorption spectra, and the Eg values were determined by the Tauc method according to the expression (1).

$$(F(R) \cdot h\upsilon)^{1/n} = A (h\upsilon - E_g) \qquad (1)$$

where *F(R)* is the Kubelka–Munk function, *h* is the Planck constant, $\upsilon$ is the photon's frequency, $E_g$ is the band gap energy, *A* is a proportionality coefficient, and *n* is a factor related to the nature of electron transition and, in this case, equal to two since anatase $TiO_2$ is an indirect band gap semiconductor [34]. The results for the Eg calculations are shown in Table 2. The Eg values for the B-$TiO_2$ 2 wt.% and B-$TiO_2$ 5 wt.% samples were similar, showing a slight blue shift in the light absorption, while the B-$TiO_2$ 3 wt.% catalyst exhibited Eg-like pure $TiO_2$ indicating very a slight displacement toward visible light, which agrees with previous studies [25,31] and the theoretical calculations made by Xu et al. [35].

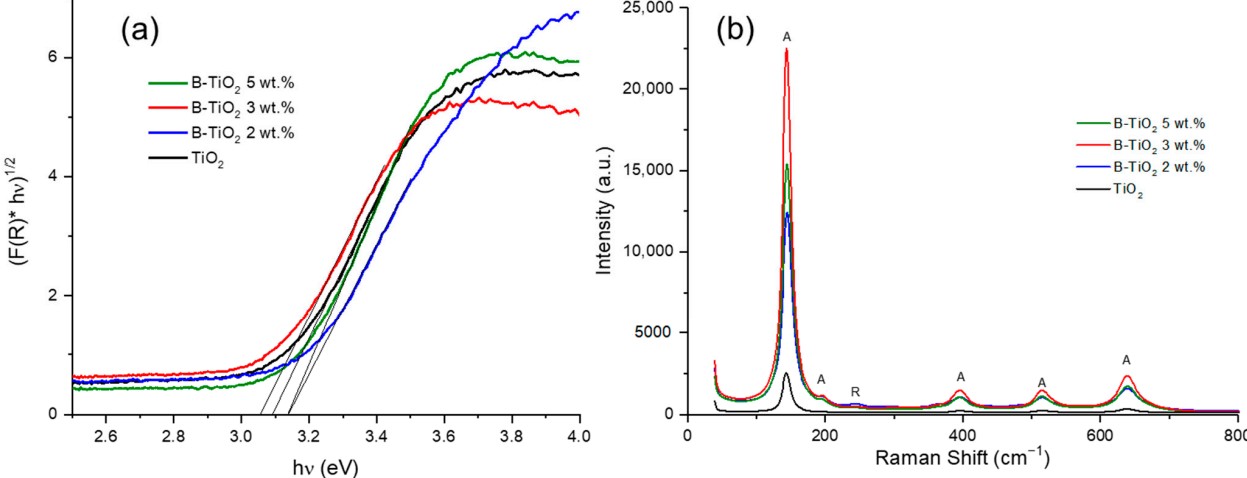

**Figure 2.** DRS spectra represented as Kubelka–Munk function and (**a**) Raman spectra of (**b**) Pure $TiO_2$ and B-$TiO_2$ with different ratios of boron.

**Table 2.** Reaction kinetic parameters using B-doped TiO$_2$ catalysts.

| Material/Catalyst Loading (g/L) | $K_{app} \times 10^{-3}$ (m$^2$/kJ) | | | R$^2$ | | |
|---|---|---|---|---|---|---|
| | 0.5 | 1.0 | 1.5 | 0.5 | 1.0 | 1.5 |
| B-TiO$_2$ 2 wt.% | 7.17 | 6.52 | 8.17 | 0.9748 | 0.9693 | 0.9578 |
| B-TiO$_2$ 3 wt.% | 7.87 | 12.69 | 9.07 | 0.944 | 0.9132 | 0.9760 |
| B-TiO$_2$ 5 wt.% | 5.09 | 5.98 | 5.59 | 0.9397 | 0.9429 | 0.9636 |

Is important to see in Table 1 that the real content of boron on TiO$_2$ (determined by the spectrophotometric method) is lower than the expected theoretical value; nonetheless, the experimental quantities are closer to the theoretical except for the 5 wt.% sample. Some studies reported similar behavior when higher amounts of boron were incorporated, e.g., in a 12 wt.% B-TiO$_2$ catalyst prepared by sol-gel, only 3.55 wt.% of B was introduced [25].

The Raman spectra of doped and non-doped TiO$_2$ are shown in Figure 2b. Raman bands corresponding to the anatase phase at 144 (Eg), 197 (Eg), 399 (B1g), 513 (A1g), 519 (B1g), and 639 cm$^{-1}$ (Eg) can be observed in all the synthesized photocatalysts. A signal at 236 cm$^{-1}$ can be seen in the B-TiO$_2$ samples and is attributed to the rutile phase of TiO$_2$ [36,37], this result agrees with the XRD analyses. With the incorporation of boron into the structure of TiO$_2$, the intensity of the signals corresponding to the crystalline phases is greater than the material synthesized without doping, confirming that the structure of the TiO$_2$ was modified.

The SEM images of the prepared photocatalysts are shown in Figure 3. The examination of the surface discloses that the morphology exhibits a spherical shape with a particle size ranging between 20 and 25 nm for the pure and doped TiO$_2$. The incorporation of boron does not seem to modify the morphology of the TiO$_2$ particles, which could be due to the small amounts of theoretical boron that were added to the catalysts. Nevertheless, the spherical particles agglomerate, generating large aggregates of particles [31].

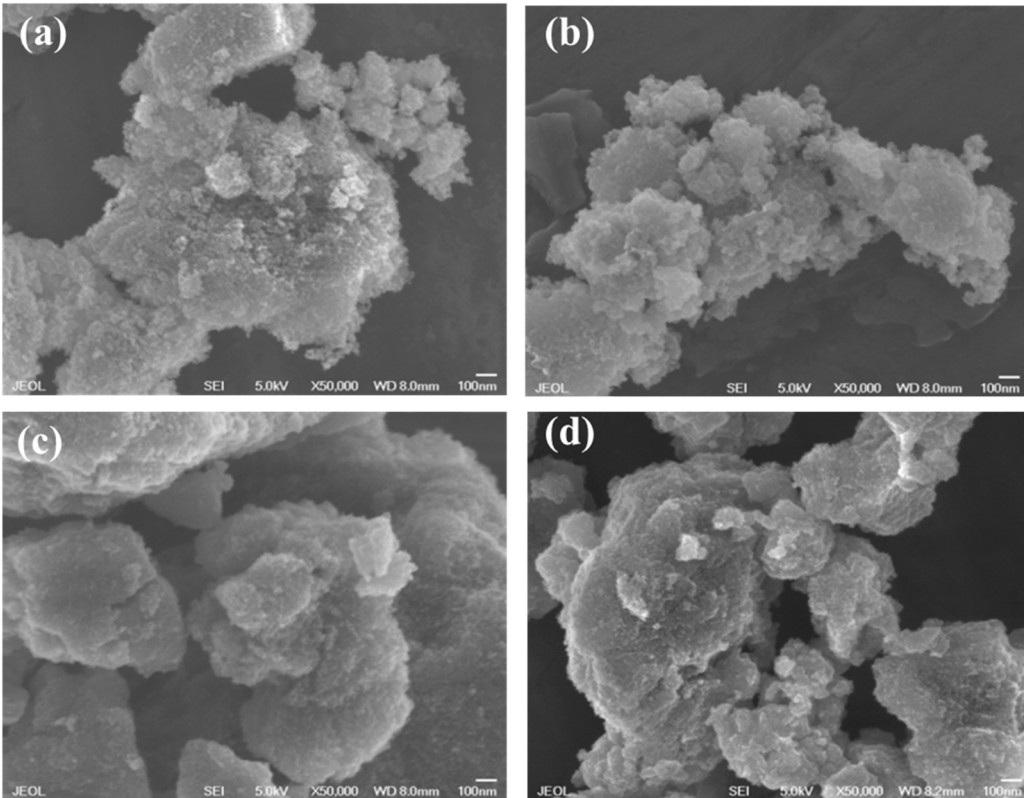

**Figure 3.** SEM images of catalysts (**a**) TiO$_2$, (**b**) B-TiO$_2$ 2 wt.%, (**c**) B-TiO$_2$ 3 wt.%, and (**d**) B-TiO$_2$ 5 wt.%.

The surface composition and valence states of elements in the samples were investigated by the XPS technique, and the survey spectra of the pure and B-doped TiO$_2$ catalysts are shown in Figure 4. In all materials, the presence of the characteristic peaks of Ti and O was observed. The small peak of C 1s is derived from amorphous carbon used for the instrument during the XPS measurement or can originate from the residual carbon of the precursor employed in the synthesis. Otherwise, the B signal at a binding energy of about 192 eV was only observed in the survey spectra of boron-doped catalysts, which is characteristic of interstitial boron on the TiO$_2$ lattice [29].

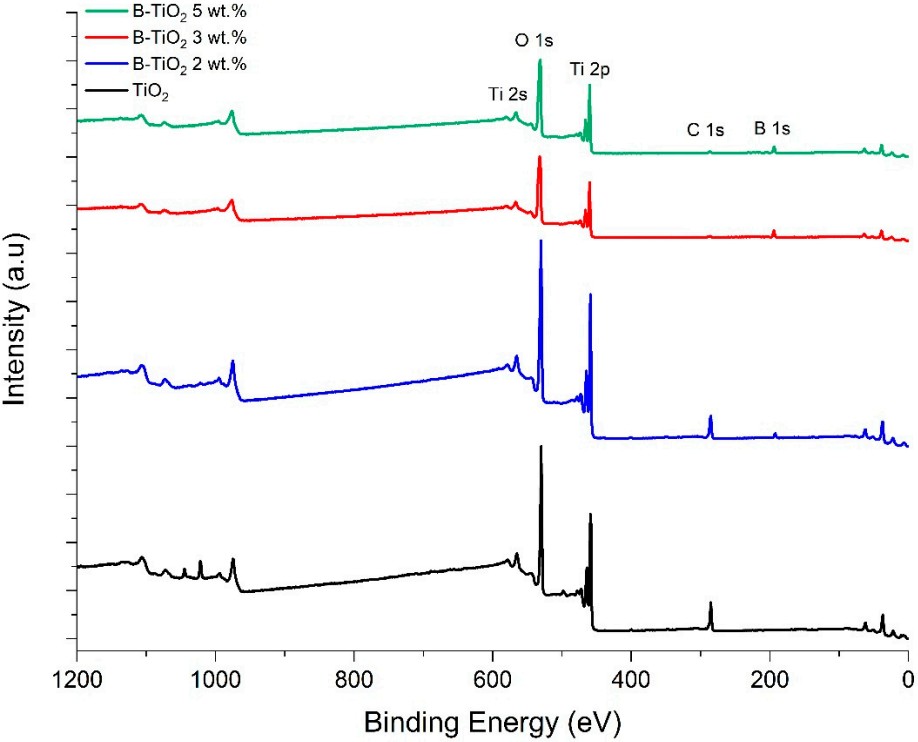

**Figure 4.** XPS survey spectra for the undoped and B-doped TiO$_2$ catalysts.

The high-resolution XPS spectra for Ti, O, and B are shown in Figure 5. The deconvolution of the Ti 2p peak from the spectra of the synthesized catalysts is depicted in Figure 5a, of which the signals of Ti 2p$_{3/2}$ at 458.28 eV and Ti 2p$_{1/2}$ at 463.98 eV corresponding to Ti$^{4+}$ are shifted to higher energies as the boron content is increased. This displacement indicates that the incorporation of boron in the TiO$_2$ lattice reduces the electron density of Ti 2p [30,31].

In Figure 5b, the O 1s spectra of the undoped and doped TiO$_2$ catalysts are shown. In the case of pure TiO$_2$, the intense peak observed at 529.48 eV is related to Ti–O bonds with a widening at higher energies that correspond to surface OH groups [25]. For the sample with 2 wt.% of boron doping (B-TiO$_2$ 2 wt.%), the O 1s region exhibited two peaks, the main Ti-O peak at 529.7 eV and a shoulder at 532 eV assigned to the B–O bonds, which confirms the incorporation of boron atoms into the TiO$_2$ lattice. Regarding the samples with 3 and 5 wt.% of boron doping (B-TiO$_2$ 3 wt.% and B-TiO$_2$ 5 wt.%), a displacement of the O 1s spectra at a higher binding energy can be observed. Additionally, the second peak at 533 eV was ascribed to the B–O bond in H$_3$BO$_3$ or B$_2$O$_3$ [25]. Therefore, the O 1s XPS analysis of these samples suggests that the boron atoms were incorporated into the TiO$_2$ matrix and some species of H$_3$BO$_3$ and B$_2$O$_3$ on the surface [30]. On the other hand, the occupation of B at interstitial positions could produce a blue shift in the absorption properties of the doped materials. This agrees with the results obtained from Eg, where no significant shift was observed.

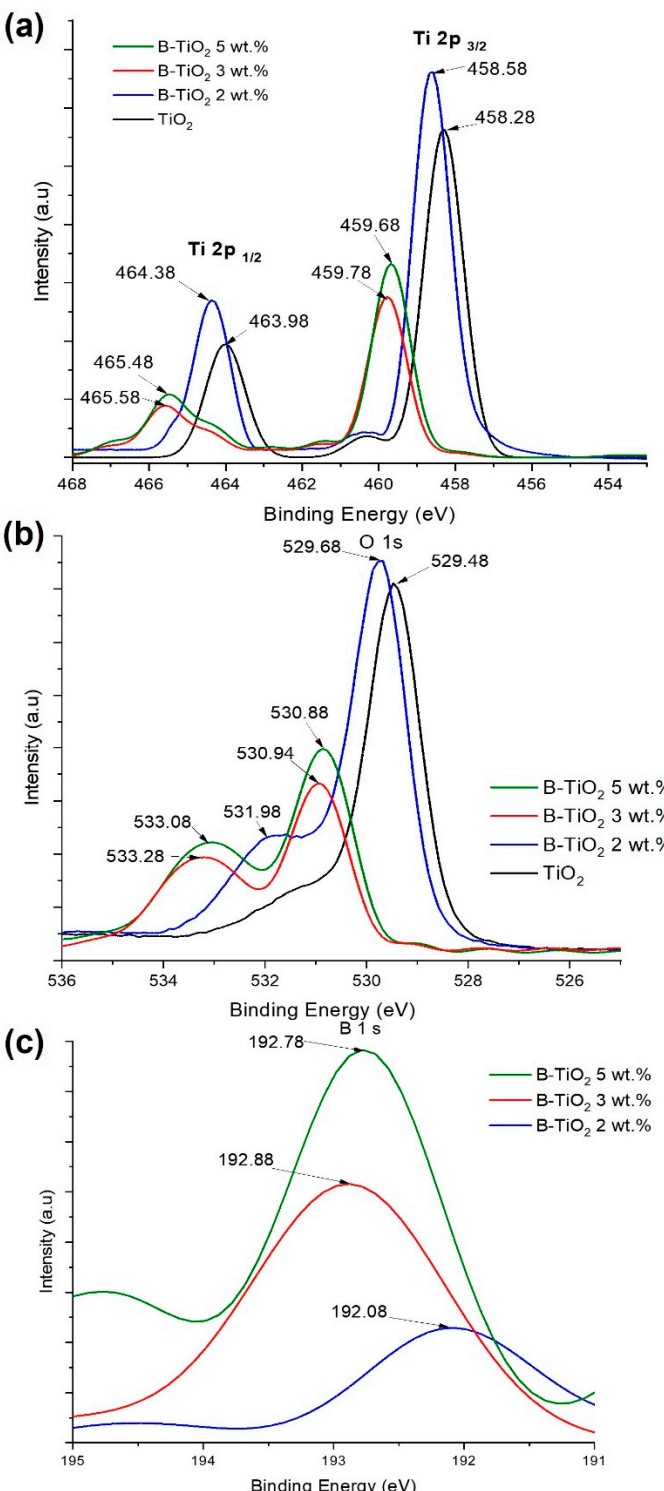

**Figure 5.** High-resolution XPS of (**a**) Ti 2p, (**b**) O 1s, and (**c**) B 1s for the B-doped catalysts.

Figure 5c shows the B 1s XPS spectra of the B-$TiO_2$ samples. The B 1s region of the B-$TiO_2$ 2 wt.% catalyst exhibits one peak at 192 eV that corresponds to the interstitial boron incorporated into the $TiO_2$ lattice, and the chemical environment surrounding it could be Ti–B–O [30,38,39].

The XPS B 1s region of the B-$TiO_2$ 3 wt.% and B-$TiO_2$ 5 wt.% nanomaterials consist of peaks at binding energies of 192.8 and 192.9 eV, respectively. Dozzi et al. [22] described that B peaks at higher binding energies (192.8, 193.3, and 194 eV) indicate the possible formation

of $B_2O_3$ microaggregates on the surface of the $TiO_2$ structure, which means that the doped catalysts with 3 and 5 wt.% of boron were covered by $B_2O_3$ dispersed on the surface in a very low content since it was not detected by XRD [30,40].

## 2.2. Photocatalytic Activity of the Prepared Catalysts

The degradation of the estriol solution under solar simulated light was conducted using different amounts of the synthesized catalysts (0.5, 1.0, and 1.5 g/L) at pH 7.2. Aliquots were sampled every 50 kJ/m$^2$ to monitor the photocatalytic process. A photolysis experiment without catalyst and adsorption tests of the B-TiO$_2$ material and in the absence of radiation were also carried out as a control test. The degradation curves of E3 by photolysis and photocatalysis using 0.5 g/L of the catalyst loading of the different prepared materials are shown in Figure 6a.

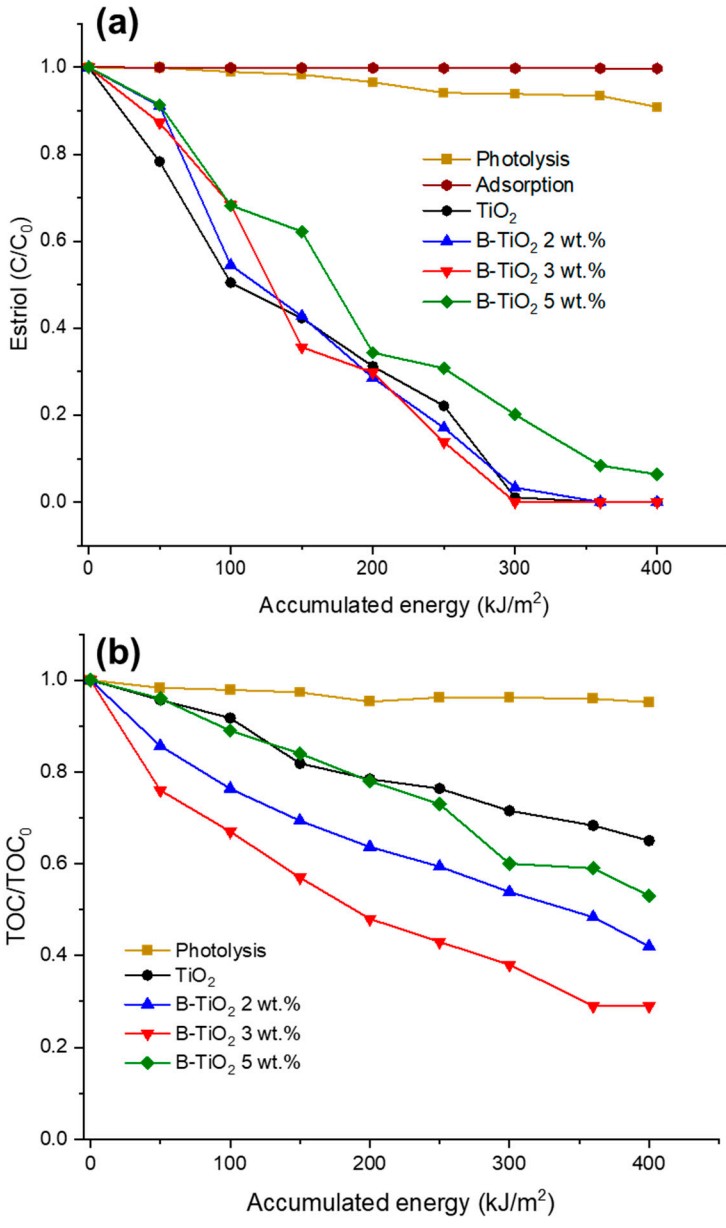

**Figure 6.** The degradation (**a**) and mineralization (**b**) curves of estriol by photolysis, adsorption, and photocatalysis with TiO$_2$ and B-doped TiO$_2$ using 0.5 g/L for the catalyst loading.

As observed, the direct photolysis was negligible for E3 degradation, and only 6.5% of the hormone at 400 kJ/m$^2$ of the accumulated energy was decomposed by solar radiation. In

addition, the test with the photocatalyst under dark conditions demonstrated that the estriol adsorption on the B-TiO$_2$ 3 wt.% material was negligible using a catalyst loading of 0.5, 1.0, and 1.5 g/L. Synthesized TiO$_2$ and all B-doped materials, except for B-TiO$_2$ 5 wt.%, achieved the complete degradation of estriol at 300 kJ/m$^2$ of the accumulated energy. Otherwise, for the solutions with the dissolved hormone treated with 1.0 and 1.5 g/L of the different modified catalysts (Figure S1), a complete degradation was obtained at 300 kJ/m$^2$ of the accumulated energy. Nevertheless, at 200 kJ/m$^2$, by using 1.0 g/L of the B-TiO$_2$ 3 wt.% sample, a higher degradation percentage (94%) was achieved. On the other hand, when B-TiO$_2$ 2 wt.% and B-TiO$_2$ 5 wt.% were used, only 77% and 69% degradation percentages, respectively, were reached (Figure S1a). In comparison, in a research work that employed a three-dimensional electrode reactor, an 80% removal efficiency of E3 was obtained [11], while in another study, complete degradation after 480 min with 0.3 Fe-TiO$_2$ under low UV radiation was accomplished [41]. Perondi et al. in 2020 conducted the degradation of E3 by UV, UV/H$_2$O$_2$, UV/TiO$_2$, and UV/O$_3$ processes. The initial concentration of E3 was 200 µg/L using TiO$_2$ under a UV light of high energy (254 nm), and less than 70% of E3 degradation was achieved in 60 min [42]. In our case, the E3 initial concentration was higher (10 mg/L), and the catalyst was irradiated by solar simulated light, which indicates the potential of B-TiO$_2$ 3 wt.% photocatalyst for the removal of estriol hormones from water effluents.

For a better understanding of the differences in the photocatalytic performance of each photocatalyst, the analysis of the apparent rate constants was conducted. In heterogeneous photocatalysis, the kinetic corresponds to a pseudo-first-order reaction according to Equation (2):

$$\ln \frac{C_0}{C} = k_{app}t \tag{2}$$

where $k_{app}$ is the apparent rate constant of the degradation reaction, $C_0$ is the initial concentration of E3, and $C$ is the concentration at time $t$. In this work, the accumulated UV energy was used as a unit of measurement for the photocatalytic reaction progress rather than time, as was reported in previous studies [43,44]. Table 2 reported the $k_{app}$ values for degradation experiments carried out with the prepared catalysts at different catalyst loading (0.5, 1.0, and 1.5 g/L). These values were calculated by plotting the data of ln (C$_0$/C) versus the accumulated UV energy (kJ/m$^2$) from the slope of the corresponding linear fit curve. The $k_{app}$ value for E3 degradation using 0.5 g/L undoped TiO$_2$ ($5.98 \times 10^{-3}$ m$^2$/kJ, R$^2$ = 0.9919) was lower compared with the boron-doped photocatalysts (Table 2). The highest degradation rate ($12.69 \times 10^{-3}$ m$^2$/kJ) was achieved with the B-TiO$_2$ 3 wt.% material when the process was conducted at 1.0 g/L.

The total organic carbon reduction was measured during the photocatalytic reaction to estimate the mineralization degree of the E3 molecule and the ability of the studied materials to destroy by oxidation the recalcitrant pollutant (Figure 6b).

Mineralization percentages of 58%, 71%, and 47% were achieved using TiO$_2$ doped with 2, 3, and 5 wt.% of B incorporated, respectively, with the use of 0.5 g/L of the catalyst added to the hormone solution. Conversely, only 35% of mineralization was attained with 0.5 g/L of pure titanium oxide. The highest mineralization was accomplished with the B-doped TiO$_2$ 3 wt.% material (Table 3).

**Table 3.** Mineralization of E3 solution (10 mg/L) with B-doped TiO$_2$ materials.

| Material/Catalyst Loading (g/L) | % Mineralization at 400 kJ/m$^2$ | | |
|---|---|---|---|
| | 0.5 | 1.0 | 1.5 |
| B-TiO$_2$ 2 wt.% | 58 | 68 | 69 |
| B-TiO$_2$ 3 wt.% | 71 | 61 | 61 |
| B-TiO$_2$ 5 wt.% | 47 | 67 | 59 |

The high mineralization degree indicates that the structure of the estriol was breakdown generating single aromatic intermediates, which were further oxidized by the •OH attack through ring-rupturing reactions into aliphatic compounds [12].

The formation of organic intermediates during the photocatalytic treatment of the E3 using B-TiO$_2$ 3 wt.% material was monitored by HPLC/EIS-MS. Based on the information provided by the MS (the *m/z* values of the ionic species) and MS data (the fragmentation profiles of the mass-selected ions), the proposed chemical structures of the reaction products are shown in Figure 7. The E3 ion peak (*m/z* 288.173) disappeared in the MS spectrum (Figure S2), suggesting that the E3 molecule was decomposed into small organic intermediates (mineralization and degradation were 71% and 94% at 400 kJ/m$^2$, respectively). The identified intermediates during the degradation of E3 were E1 and E2, which were subsequently degraded to form the compound of *m/z* 256.146 [45], *m/z* 230.131, *m/z* 166.063 (2-ethyl-4-hydroxybenzoic acid), and *m/z* 126.032 (hydroxyquinol), resulting in ring-opened products [11,46].

**Figure 7.** Proposed fragmentation pathways of E3 in the presence of B-doped TiO$_2$ 3 wt.% catalyst under solar simulated light.

Although short-chain carboxylic acids such as oxalic, oxamic, acetic, and formic are generally described as the further oxidation of the intermediate products formed, oxalic acid was identified by ion-exclusion chromatography in this study. This compound is considered highly recalcitrant to oxidation [47], and therefore, it remained in the final solution.

The enhancement of the degradation and mineralization of E3 using the B-TiO$_2$ 3 wt.% catalyst can be ascribed to the high surface area (153.2 m$^2$/g), interstitial incorporation of B into the TiO$_2$ lattice, the formation of a small amount of B$_2$O$_3$, and the presence of the rutile phase which could promote the separation of the photogenerated electrons and holes for efficient redox reactions [22]. The formation of rutile can improve the charge in the carrier's separation because of the valence (VB) and conduction band (CB) alignment between the anatase and rutile phases. The electrons can be transferred from rutile CB to anatase CB and the holes from anatase VB to rutile. In addition, the B incorporated interstitially acts as a trap of electrons to extend the life of the photogenerated $e^-/h^+$ pairs and decreases the recombination rate. Therefore, the electrons and holes react with adsorbed water and oxygen, generating oxidant species (mainly hydroxyl radicals) to efficiently oxidize the E3 molecule [27].

Otherwise, the toxicological assessment of the effluent after the photodegradation of E3 is essential because it has been reported that more toxic by-products than the original molecules can be formed at the end of the pollutant treatment [43,48]. Toxicity tests using the *V. fischeri* bacterium have been widely reported as an effective method for evaluating toxicity in the degradation of organic pollutants by heterogeneous photocatalysis [49,50]. In this context, Figure 8 depicts the results obtained during the toxicity evaluation of the photocatalytic degradation of estriol.

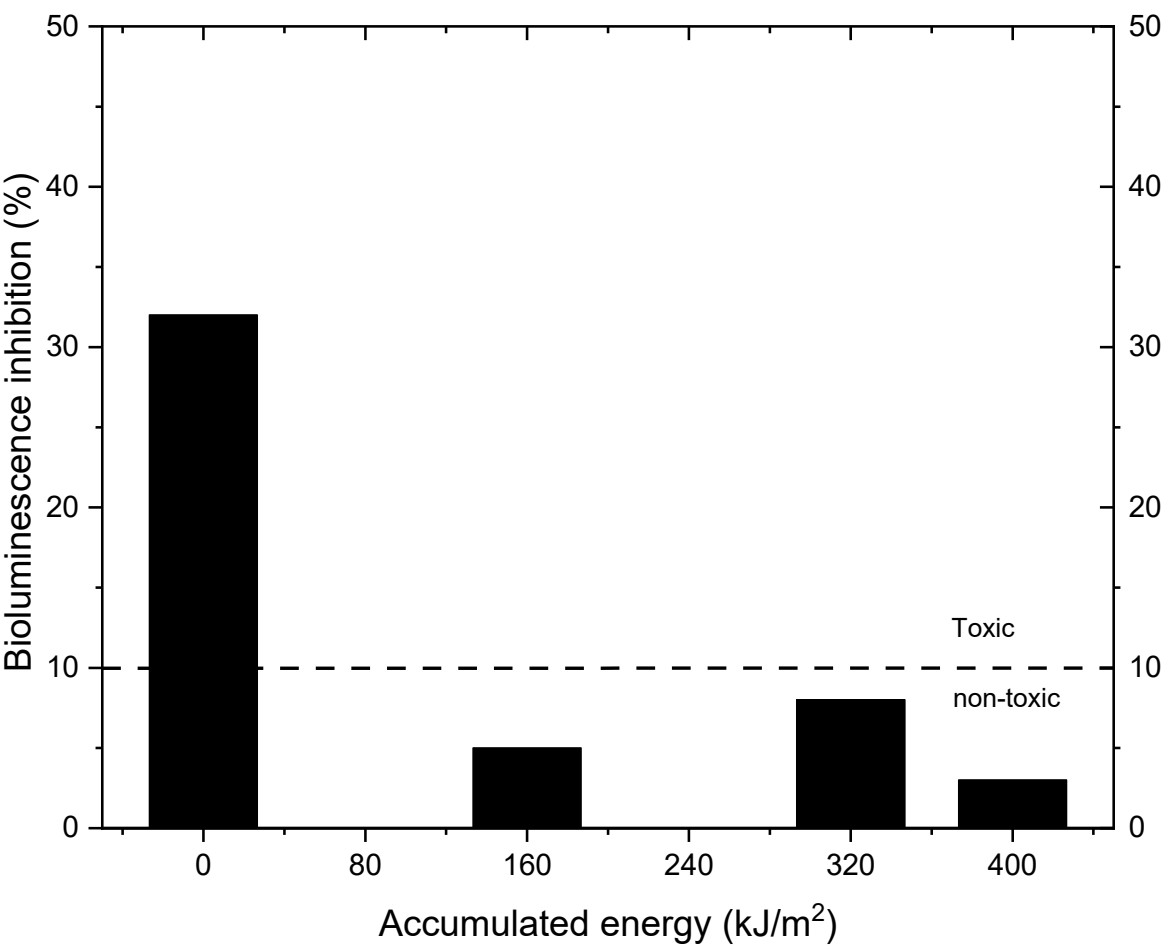

**Figure 8.** Evolution of the toxicity during photocatalytic treatment of E3 using the B-TiO$_2$ 3wt.% catalyst, expressed as the luminescence inhibition (%) of the bacteria *Vibrio fischeri*.

The test revealed an initial inhibition of 32% of *V. fischeri* bioluminescence (0 kJ/m$^2$ accumulated energy), indicating that the initial solution with estriol was toxic. According to Mexican regulations (NMX-AA-112-SCFI-2017) [51], a bioluminescence inhibition higher

than 10% in contrast to the control sample indicates that the sample is toxic; samples with less than 10% inhibition are categorized as non-toxic. The bioluminescence inhibition values during the E3 degradation decreased to 5, 8, and 3% at 160, 320, and 400 kJ/m$^2$ of the accumulated energy, respectively, using the B-TiO$_2$ 3 wt.% catalyst, demonstrating that a non-toxic solution is attained from 160 kJ/m$^2$ of accumulated energy.

## 3. Material and Methods

### 3.1. Chemicals

All chemicals used were analytical-grade reagents. Estriol (98%) and 2-butanol ($\geq$99.8%), titanium (IV) butoxide (97%), acetonitrile (99.8%), and glacial acetic acid ($\geq$99.7%), boric acid ($\geq$99.5%) were purchased from Sigma-Aldrich Co. (Burlington, MA, USA). Deionized water was used in all the experiments.

### 3.2. Catalyst Preparation

The sol-gel method was used to synthesize TiO$_2$ [52] and B-doped TiO$_2$. Titanium butoxide (15.13 mL) and a mixture of 2-butanol (79.94 mL), and the desired amount of boric acid solution were mixed under constant stirring. The glacial acetic acid acted as a chelating agent to control the hydrolysis and was added until reaching a pH of 3.0. Subsequently, deionized water (7.76 mL) was added dropwise until the sol was obtained. The gel was formed when the mixture was hydrolyzed at room temperature for 1 h under vigorous stirring. Finally, the obtained gel was aged for 24 h at room temperature and dried at 80 °C in an oven for 12 h to remove the solvents and then calcined at 500 °C in a muffle furnace for 1 h, using a heating rate of 2 °C/min. The calculated boron concentrations were 2, 3, and 5 wt.%, respectively. Pure TiO$_2$ powder was synthesized with the same process without the incorporation of boric acid.

### 3.3. Characterization

Diffuse UV-Vis reflectance spectroscopy (DRS), was applied using a UV-Vis spectrometer Nicolet 300 equipped with an integrating sphere, Thermo scientific, Waltham, MA, USA. Scans were performed in a range from 200 to 800 nm. The powder X-ray diffraction (XRD) analyses were carried out at room temperature using a D2 Phaser, Bruker X-ray diffractometer, Bruker, Germany with Cu K$\alpha$ radiation ($\lambda$ = 1.5418 Å) measured in a 2$\theta$ range between 5 and 90° to identify the crystalline phase of the obtained materials. Crystallite sizes were calculated from the line broadening of the main anatase X-ray diffraction peak of higher intensity (25.5°) by the Scherrer equation. SEM images were obtained with a JSM-6701F JEOL, Japan scanning electron microscopy working at 200 kV to observe the morphology of the samples.

The elemental composition of the catalysts was investigated by X-ray photoelectron spectroscopy (XPS) using K-Alpha instrument, Thermo Scientific (Waltham, MA, USA) with a monochromatic X-ray source (Al K$\alpha$-14.5 kW). All measurements were made in an ultra-high vacuum chamber pressure of $10^{-6}$ Torr. The specific surface area and porosity of the catalysts were determined at 77 K by analyzing N$_2$ adsorption isotherms using the Autosorb-1 Automated Gas Sorption instrument, Quantachrome, Boynton Beach, FL, USA. The crystalline structure of the materials was analyzed by Raman spectroscopy using a triple grating Jobin-Yvon T64000 spectrometer, Horiba, Japan.

The real amounts of boron incorporated into the TiO$_2$ synthesized materials were quantified using the carminic acid assay reported by Floquet et al. [53], using a Cary 50 Scan UV-Vis spectrophotometer, Varian, Australia at 610 nm. To obtain the solid dissolved in a liquid solution, 0.1 g of each material was digested using a mixture of H$_2$O$_2$ and HCl at 180 °C for 20 min in a MARS 6 microwave oven, CEM, Charlotte, NC, USA.

### 3.4. Photocatalysts Performance

The photocatalytic activity under simulated solar radiation of TiO$_2$ and B-TiO$_2$ powders was evaluated by the degradation rates of estriol (10 mg/L) in an aqueous solution

(250 mL) at pH 7.2, containing 0.5, 1.0, and 1.5 g/L of catalyst, respectively. The reaction mixture was ultrasonically dispersed for 30 min in darkness to establish an adsorption–desorption equilibrium between the solution and the catalyst and was then irradiated by a Suntest XLS+ , Atlas, Chicago, IL, USA solar simulator that uses a xenon lamp equipped with a daylight filter that emits radiation from 300 to 800 nm. The solar simulator was set to emit an instantaneous UV radiation of 60 W/m$^2$. The degradation reaction was monitored at different values of the accumulated UV energy (QUV) (kJ/m$^2$) under the established conditions, and 50 kJ/m$^2$ accumulated in 15 min. The collected samples were filtered through 0.45 μm nylon filters to remove catalyst particles.

### 3.5. Analytical Measurements

The concentration of estriol was analyzed by High-Performance Liquid Chromatography (YL9100 Young Lin, Korea), equipped with a diode array detector (DAD) and a HyperClone C18 column. The mobile phase was a mixture of acetonitrile and water with a ratio of 35:65. The flow rate was 1 mL/min, the injection volume was 20 μL, and the detection wavelength was 200 nm. The total organic carbon (TOC) was measured using a TOC Analyzer Equipment TOC-V CSH, Shimadzu, Japan. To carry out this analysis: 14 mL aliquots were taken, filtered, and analyzed.

The by-products from the photocatalytic degradation were analyzed by HPLC and mass spectrometry (MS) with an electrospray ionization (ESI) source (Compact Mass Spectrometer (CMS), Advion, Ithaca, NY, USA, operating in the positive ion mode. The chromatographic separation was performed in the HPLC system YL9100 (Young Lin), an automatic injector, and C-18 HyperClone (250 mm × 4.6 mm, i.d. 5 μm) column (Phenomenex, Torrance, CA, USA).The mobile phase eluents were acetonitrile: water (50:50). The injection volume was 20 μL, and the flow rate was 0.3 mL/min. The reaction aliquots were directly analyzed in HPLC/ESI-MS, the ion source at a flow rate of 0.30 mL/min, and the related mass spectra were obtained as an average of 50 scans, each requiring 0.02 s. In the full scan mode, *m/z* 10, at 600. The typical ESI conditions were as follows: heated capillary temperature, 200 °C, sheath gas (N$_2$) at a flow rate of 20 mL/min, spray voltage −2.60 kV, nebulizing gas, and curtain gas flows, which were at instrument settings of 30, and 10 mL/min, respectively.

Generated carboxylic acids were detected by ion-exclusion chromatography using a Bio-Rad Aminex HPX-87H column (300 mm × 7.8 mm) with UV detection (200 nm) using 4 mM H$_3$PO$_4$ mobile phase at 0.8 mL/min.

### 3.6. Toxicity Assessment

The bioluminescence inhibition in *Vibrio fischeri* (*V. fischeri*) bacteria was evaluated using the manufacturer's procedure for low-toxicity samples on the DeltaTox® II toxicity analyzer, Modern Water, New Castle, DE, USA. The bioluminescence of *V. fischeri* decreases in the presence of pollutants, and toxicity was expressed as a function of the luminescence inhibition percentage. The toxicity of estriol was evaluated using the B-TiO$_2$ catalyst that exhibited the best photocatalytic performance, and samples were collected at different accumulated energy values during the reaction. The light output of the samples was compared to the light output of a control after 10 min of exposure to the bacteria (reagent blank).

### 4. Conclusions

The catalytic efficiency of the prepared catalysts indicated that the sample with 3 wt.% of boron incorporation exhibited the best performance on the degradation and mineralization of estriol, achieving its complete degradation and 71% mineralization at 400 kJ/m$^2$ of accumulated energy. The incorporation of the non-metal induces the formation of small amounts of the rutile phase of TiO$_2$, increases the specific surface area, and reduces the crystallite size. The Eg values of the B-doped catalysts were similar to pure TiO$_2$; however, the improvement in the photocatalytic activity of titania doped with B was attributable to a lower recombination rate of $e^-/h^+$ pairs. The boron-doped TiO$_2$ catalyst confirms its

potential to degrade organic pollutants, such as the steroid hormone estriol, obtaining a non-toxic effluent using solar light as an irradiation source.

**Supplementary Materials:** The following supporting information can be downloaded at: https://www.mdpi.com/article/10.3390/catal13010043/s1, Figure S1: Degradation and mi-neralization curves of estriol by photocatalysis with B-doped $TiO_2$ using 1.0 g/L (a) and (c), and 1.5 g/L (b) and (d) of catalyst loading; Figure S2: Mass spectrum of E3 degradation at different accumulated energy values: (A) 0, (B) 200, (C) 300, and (D) 400 kJ/m$^2$.

**Author Contributions:** L.Y.R.-Q.: Conceptualization, Data curation, Methodology, Formal analysis, Writing—original draft. D.P.-S.: Visualization, Writing—review and editing. J.L.G.-M.: Writing—review and editing, J.C.M.-S.: Writing—review and editing, E.J.R.-R.: Investigation, Methodology, Writing—review and editing. A.H.-R.: Funding acquisition, Investigation, Supervision, Methodology, Writing—original draft. All authors have read and agreed to the published version of the manuscript.

**Funding:** This research was founded by CONACyT, project 181057.

**Data Availability Statement:** Almost all the data generated or analyzed during this study are included in this published article and its Supplementary Information file. Additional data are available from the corresponding author (aracely.hernandezrm@uanl.edu.mx).

**Acknowledgments:** The authors gratefully acknowledge financial support from the National Council of Science and Technology, Mexico (CONACyT, project 181057).

**Conflicts of Interest:** The authors declare no conflict of interest.

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
