# Peer review of "Photocatalytic Degradation and Mineralization of Estriol (E3) Hormone Using Boron-Doped TiO2 Catalyst"

_catalysts, doi:10.3390/catal13010043_

Round 1
Reviewer 1 Report
The work “Photocatalytic Degradation and Mineralization of Estriol (E3) 2 Hormone Using Boron-Doped TiO2 Catalyst” is reported. The work is recommended for major revision.
1. Boron doped TiO2 has been reported in many studies. This is not a novel idea. A strong justification should be provided in the introduction section.
2. This study was conducted at a very low concentration of estriol and at a very high catalyst dose. Thus, the removal of estriol only due to the adsorption (in dark) should be provided.
3. The bandgap values reported in table 1 seem wrong. As per figure 2, bandgap values are not the same. That should be corrected.
4. In tables 2 and 3, add kinetic data for the reaction with bare TiO2.
5. Compare the obtained results for estriol degradation with the earlier studies.
Author Response
Reviewer 1
The work “Photocatalytic Degradation and Mineralization of Estriol (E3) 2 Hormone Using Boron-Doped TiO2 Catalyst” is reported. The work is recommended for major revision.
- Boron doped TiO2 has been reported in many studies. This is not a novel idea. A strong justification should be provided in the introduction section.
R/Thank you for your observation we extend this paragraph:
In a B-TiO2 doped catalyst, the content of boron, the TiO2 precursor, and the calcination temperature determines their structural, optical, and textural properties, and these in turn affect the photocatalytic behavior. Thus, the effect of TiO2 boron doping still deserves to be further investigated since its behavior also depends on the used radiation source because different results have been obtained regarding to the contribution of B on the TiO2 bandgap energy. Additionally, until our knowledge, the doped B-TiO2 material has not been explored on the degradation reaction of the E3 hormone in aqueous media.
- This study was conducted at a very low concentration of estriol and at a very high catalyst dose. Thus, the removal of estriol only due to the adsorption (in dark) should be provided.
R/ Thank you for your suggestion, adsorption test of the materials under dark conditions for 2 h was conducted for the E3 solution (10 mg/L) and the results were added in the manuscript and shown in Figure 6a).
- The bandgap values reported in table 1 seem wrong. As per figure 2, bandgap values are not the same. That should be corrected.
R/ Bandgap values were recalculated and correspond to those reported in Table 1. A paragraph was added clarifying how these values were calculated in the manuscript, and in Figure 2a) solid lines were added on the slope to better observe the intercept with the x-axis.
- In tables 2 and 3, add kinetic data for the reaction with bare TiO2.
R/ Thank you for your suggestion, however, since the kinetic constant and the percentage of mineralization achieved with bare TiO2 were only measured at a fixed catalyst loading (0.5 g/L), we decided to report these values within the text.
- Compare the obtained results for estriol degradation with the earlier studies.
As we remark in the introduction section, we did not find studies regarding to the degradation of E3 by Boron-doped TiO2 photocatalysts. Only a few studies have been reported on E3 oxidation by an electrochemical method, Ozone or photocatalysis using TiO2 under UV radiation. The comparison with our results was added to the manuscript.
Reviewer 2 Report
In this manuscript, the authors described the synthesis of boron doped TiO2 catalysts by the sol-gel method and their application in degradation and mineralization of estriol hormone. The optimal ratio of boron in catalyst was explored for efficient decomposition of estriol. I think it can be accepted for publication after minor revision. The main concerns are as follows:
1. TEM should be performed to show the morphology and microstructure of the catalysts.
2. In Fig. 5, the chemical shift of characteristic peaks should be marked clearly.
3. The mechanism of boron doped TiO2 catalyst was not explained clearly in this manuscript.
Author Response
Reviewer 2
In this manuscript, the authors described the synthesis of boron doped TiO2 catalysts by the sol-gel method and their application in degradation and mineralization of estriol hormone. The optimal ratio of boron in catalyst was explored for efficient decomposition of estriol. I think it can be accepted for publication after minor revision. The main concerns are as follows:
- TEM should be performed to show the morphology and microstructure of the catalysts.
R/ The reviewer is right, this technique is of great importance to analyze the morphology and microstructure of the materials. However, we do not have access to this instrument in our university. In some cases, we use to send the samples with colleagues out of our city, however for the short time for the revision (ten days) was not possible to carry out this study. We apologize for the inconvenience.
- In Fig. 5, the chemical shift of characteristic peaks should be marked clearly.
R/ In Figure 5, the binding energies for each of the signals were added in order to clearly show the shifting of the characteristic peaks.
- The mechanism of boron-doped TiO2catalyst was not explained clearly in this manuscript.
R/ Thank you for your suggestion, an explanation was included on page 14, lines 389-395.
Reviewer 3 Report
The draft presented B-TiO2 (with different amounts of Boron) as promoter photoheterogeneous catalyst for degradation of Estriol (E3) Hormone under exposed light source. There is some issues related to the work.
In the preparation catalytic section there is no references were mentioned, which counter with what mentioned in the introduction that several group prepared the catalyst with sol-gel. Is this preparation setup new?
According to the presented results in the draft the TiO2 as itself work very well for the degradation of the substrate (E3) without the modification with B heteroatom. Also, in the XRD no diffraction lines corresponding to B oxides and missing the at% on the surface from xps. I recommended you to make EDX.
More evidence needed for the role of B in the system, due to the Characterization part and the small amounts of the B and highly effective TiO2 parent.
Any details: analysis information’s about the degradation products such as HPLC ,GC/MS …so on, can increase the value of the work.
Author Response
Reviewer 3
The draft presented B-TiO2 (with different amounts of Boron) as promoter photoheterogeneous catalyst for degradation of Estriol (E3) Hormone under exposed light source. There is some issues related to the work.
1-In the preparation catalytic section there is no references were mentioned, which counter with what mentioned in the introduction that several group prepared the catalyst with sol-gel. Is this preparation setup new?
R/ The synthesis method of TiO2 by sol-gel technique was implemented before in our research group and reported in the article Catalysis Today 209, 15 June 2013, 35-40. In this work, we only incorporated boron during the TiO2 synthesis by using H3BO3 as the precursor. Our reference was added to the methodology section.
2.-According to the presented results in the draft the TiO2 as itself work very well for the degradation of the substrate (E3) without the modification with B heteroatom. Also, in the XRD no diffraction lines corresponding to B oxides and missing the at% on the surface from xps. I recommended you to make EDX.
R/ Thank you for your recommendation, since the XPS technique showed that B was found in our catalyst and the point measurement of the wt.% incorporated by this technique is not entirely accurate, and due to we do not have access to conduct an EDS analysis, we decided to measure B content by a spectrometric method, after digestion in a microwave oven of the materials. The wt.% of B incorporated in the materials was added in table 1. In addition, the methodology and its results about it were added to the manuscript.
3-More evidence needed for the role of B in the system, due to the Characterization part and the small amounts of the B and highly effective TiO2 parent.
Boron was mainly incorporated interstitially, and it acts as a trap of electrons to extend the life of the photo-generated e-/h+ pairs decreasing the recombination rate. Therefore, the electrons and holes react with adsorbed water and oxygen-generating oxidant species (mainly hydroxyl radicals) to efficiently oxidize the E3 molecule achieving a high degree of mineralization and producing non-toxic byproducts in the final solution.
4. Any details: analysis information’s about the degradation products such as HPLC ,GC/MS …so on, can increase the value of the work.
R/ Thank you for your suggestion. The determination of intermediates during the photocatalytic degradation of E3 was performed using HPLC-MS and ion-exclusion chromatography. In addition, a fragmentation pathway of E3 in the presence of a B-doped TiO2 3 wt.% catalyst under simulated solar light was proposed. Figures 7 and S2 were added, and the methodology and results of this section were added to the manuscript.
Round 2
Reviewer 1 Report
None
Reviewer 3 Report
The manuscript does not need further modifications